

# Measurements of the gingival papillae architecture using cone-beam computed tomography in young Chinese adults

Gang Yang[1], Jie Cao[1], Wenjie Hu[1] and Kwok-Hung Chung[2]

[1] Department of Periodontology, Peking University School and Hospital of Stomatology, National Clinical Research Center for Oral Disease, National Engineering Laboratory for Digital and Material Technology of Stomatology, Beijing Key Laboratory of Digital Stomatology, Beijing, China

[2] Department of Restorative Dentistry, University of Washington, Seattle, WA, United States of America

## ABSTRACT

**Background**. The aim of this study was to measure the morphology of interdental papillae (IP) in maxillary anterior teeth using cone-beam computed tomography (CBCT).

**Methods**. Twenty-seven periodontally healthy subjects with 135 IP were evaluated by means of periodontal examination and a CBCT scan with an elastomeric matrix containing radiopaque material in position. According to the status of tooth contact and presence of IP, subjects were categorized into three groups: open contact point, complete papillae, and deficient papillae group. The papillae height (PH), facial-lingual thickness (FLT), and interdental distance (IDD) were measured. Data was analyzed with the significance level at $\alpha = 0.05$.

**Results**. The mean PH values were $4.17 \pm 0.51$ mm, $3.99 \pm 0.61$ mm, and $3.99 \pm 0.62$ mm, for the open contact group, complete papilla, and deficient papilla group, respectively. The PH values of the recorded sites among central incisors, lateral incisors, and canine were $4.13 \pm 0.56$ mm, $3.87 \pm 0.63$ mm, and $4.07 \pm 0.58$ mm, respectively. No statistically significant differences of the PH values were determined between the above three tested groups as well as between the different sites ($p > 0.05$). The IDD values obtained from the complete papilla group were significantly lower than the other tested groups ($p < 0.05$).

**Conclusion**. The contact point condition of upper anterior sextants may not influence the presence and dimension of the IP in periodontally healthy subjects.

Corresponding author
Wenjie Hu, huwen-jie@pkuss.bjmu.edu.cn

## INTRODUCTION

Interdental papilla (IP), which occupies the interdental embrasure, not only acts as a barrier in protecting the periodontal structures, but also plays a critical role in facial esthetics (*Chow et al., 2010*; *Chen et al., 2010*; *Kim et al., 2011*). Preserving the gingival tissue in the embrasures of the maxillary anterior sextant is a key consideration in periodontal, restorative, and implant treatment (*Tarnow, Magner & Fletcher, 1992*; *Kolte, Kolte & Mishra, 2014*). Various factors may influence the presence or absence of the IP including the distance between underlying bone crest (BC) and contact point (CP),

interdental distance (IDD), periodontal phenotype, and adjacent tooth form (*Tarnow, Magner & Fletcher, 1992*; *Cho et al., 2006*; *Chen et al., 2010*; *Chow et al., 2010*; *Kim et al., 2011*). Among these factors, the CP-BC distance has been considered as a dominant factor. According to the previous report, the papilla was often completely present when the CP-BC distance was ≤ 5 mm (*Tarnow, Magner & Fletcher, 1992*). Moreover, it has been reported that the positioning of proximal contact coronally during prosthetic treatment may cause loss of the IP (*Nozawa et al., 2011*). Previous studies indicate that the IDD is a crucial determinant of PH (*Cho et al., 2006*; *Kim et al., 2011*). The facial-lingual thickness (FLT) of the base of each papilla was suggested as a potential factor influencing the presence of the IP (*Gastaldo, Cury & Sendyk, 2004*; *Chang, 2008*). Previously, there were several studies reported the relationship between the presence and the dimensional size of the gingival architecture using conventional techniques (*Chow et al., 2010*; *De Santana, De Miranda & De Santana, 2017*).

Cone-beam computed tomography (CBCT) has been widely used in clinical examination and evaluation of the hard tissue in periodontal and implant treatment (*Scarfe, Farman & Sukovic, 2006*; *Vera et al., 2012*). It has been reported that one of the shortcomings of CBCT is its inability to discriminate soft tissues that has rendered CBCT an exclusive tool for hard tissue imaging (*Guerrero et al., 2006*). The soft tissues of the lips and cheeks collapse on the facial gingiva and the tongue occupies the space of the space of the oral cavity, thus completely preventing the visualization of the soft tissues of the periodontium (*Januario, Barriviera & Duarte, 2008*). In fact, CBCT can be used to measure the dimensions of gingival tissue when soft tissue retraction or impression material with opaque agent is applied to separate the gingiva and other soft tissues (*Januario, Barriviera & Duarte, 2008*; *Cao et al., 2015*). Thus, the CBCT imaging can visualize and measure both hard and soft tissues of periodontium and dentogingival attachment apparatus accurately. The clinicians will also be capable of determining the relationships among structures of periodontium such as the gingival margin, bone crest, and the cementoenamel junction, as well as measuring the width of facial and palatal/lingual alveolar bone and gingiva. This will aid clinicians in the planning and execution of periodontal and implant procedures with increased predictability. The dimension of facial and palatal/lingual alveolar bone and gingiva had been measured by previous study (*Januario, Barriviera & Duarte, 2008*; *Cao et al., 2015*; *Ogawa et al., 2020*), while the architecture of the IP has not been visualized and measured by CBCT imaging yet. The aim of this investigation was to assess the spatial architecture of the IP in the periodontally healthy young Chinese adults using CBCT. The null hypothesis was that no significant correlation would be found among different tooth contact types and presence of IP.

## MATERIALS & METHODS

### Sample selection

This study was approved by the Biomedical Ethics Committee of Peking University School and Hospital of Stomatology (approval ID PKUSSIRB-2012047) and was conducted in accordance with the Helsinki Declaration of 1975, as revised in 2013. Written informed

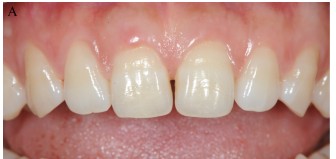
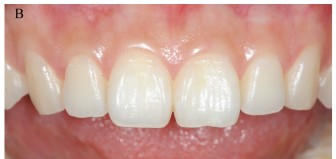
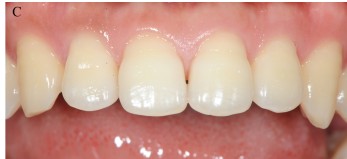

**Figure 1** **Representative maxillary sextant group.** (A) No contact point; (B) Complete papilla; (C) Deficient papilla.

consent was obtained from the participants in accordance with the guidelines of the committee for the subject selection process. The recruitment of research subjects was conducted and enrolled among 486 dental students and staff under preventive care screened at the School of Stomatology. The selection criteria included, (i) Chinese adults of 20 years or older with fully erupted permanent dentitions, (ii) no medications known to increase the risk of gingival hyperplasia, (iii) no pregnancy, (iv) grossly healthy maxillary anterior gingival tissue with probing depths $\leq 3$ mm and gingival index of grades 0 to 1, (v) well-aligned maxillary anterior teeth, (vi) absence of gingival recession, (vii) no previous surgical periodontal treatments in the anterior maxillary region, (viii) no artificial crowns and anterior restorations, (ix) no missing teeth in the maxillary anterior region. Twenty-seven periodontally healthy subjects including 12 males and 15 females were recruited with the mean age of 24.5 years old. All subjects were treated with supra-gingival scaling in the Department of Periodontics 2 weeks before clinical examination and the CBCT imaging.

## Clinical examination

All proximal contacts and spaces between the right and left maxillary canine were examined clinically using dental floss (Oral-B) and explorer (Hu-Friedy) clinically. Each papilla was given a papilla score (PS) of 0-3 based upon Nordland and Tarnow's classification system (*Nordland & Tarnow, 1998*). The papilla was defined as complete (PS = 0) or deficient (PS $\geq$ 1). The open contact case was defined with no resistance by passing a dental floss through the height of contours of two adjacent teeth. There are three types of the relationship between CP and IP categorized and grouped as the open contact area group, the deficient papilla group, and the complete papilla group (Fig. 1).

## CBCT examination

The CBCT scanning procedure was performed as previously described (*Cao et al., 2015*). Briefly, a matrix, including mesial sides of the maxillary first premolars, was fabricated in the mouth individually using silicone impression material (RAPID putty, Coltène/Whaledent AG). The thickness of the silicone matrix was approximately 6 mm, with at least 8 mm of silicone material extending apically from the free gingival margins facially and palatally. After a 4-minute initial set intraorally, the silicone matrix was removed from the patient's mouth, and the intaglio surface was trimmed evenly with a scraper to remove a layer approximate 1 mm of silicone material from the surface opposing the mucogingival region of the matrix in order to create approximately 1 mm of relief space. The putty-wash

impression technique was applied, and the final impression was made using a mixture of barium sulfate powder (Reagent grade, Qingdao Dongfeng Chemical) and alginate impression material (Heraeus Kulzer) in a ratio of 1:2 by weight. It was then loaded into the previously prepared matrix to capture the detail of the mucogingival tissue contour. The impression was allowed an intraoral setting time of 3 min. With the impression material with opaque agent loaded matrix in place, a scan of the maxillary arch and dentition was obtained with the NewTom CBCT machine, in the Radiology Department, Peking University School and Hospital of Stomatology. All the scans were taken with 0.125 mm slice thickness, 8 × 8 cm field of view size and 360 degrees with the same imaging unit (NewTom) and pixel size of 0.125 mm, exposure parameters: 15 s, 110 kVp, and 17 mA. The DICOM (Digital Imaging and Communications in Medicine) data was generated and transferred to a volumetric imaging software (Mimics 17.0, Materialise) in which three-dimensional reconstruction was conducted and the image analyses were carried out.

## Examination of the morphology of IP

Twenty-seven periodontally healthy subjects consisting of 135 maxillary anterior papillae were enrolled in this study. The silicone matrix with opaque agent on the surface of gingival and mucogingival tissue appeared radiopaque in imaging software (Fig. 1). The cross cursor was located in the middle of the two adjacent teeth of the papilla to be measured. The horizontal axis was positioned across the center of two adjacent root canals in the axial plane. The vertical axis was positioned across the contact area or the top of contours for the open contact cases between two teeth in the coronal plane. In the sagittal plane, the vertical axis was positioned parallel to the line connecting center point of two adjacent root canals. The linear distance between the cortical bone and the most coronal point of the inner radiopaque surface parallel to the vertical axis represented PH. The linear distance between the facial and palatal inner radiopaque surface across the alveolar bone crest represented the FLT of the papilla. The IDD was measured as the distance between the two adjacent roots along the horizontal axis which across the center of the adjacent root canals in axial plane. All imaging measurements were obtained by a single examiner (GY). In addition, 10 CBCT imaging measurements were repeated 1 week later by the examiner in order to test the intra-examiner reliability.

## Statistical analysis

Data expressed as means ± standard deviation (SD) were analyzed with the statistical software package (IBM, SPSS Statistics 20.0, Chicago, IL). The intra-examiner reliability was tested by intra-class correlation coefficient (ICC). The values of the PH, FLT, and IDD measurements among the open contact point group, complete papilla group, and deficient papilla groups were tested using 1-way ANVOA and Tukey tests with the significance level at $\alpha = 0.05$.

## RESULTS

The distribution of the proximal contact of the included subjects are listed in Table 1. The ICC values obtained range from 0.843 to 0.985. Two morphological forms of the

**Table 1   Results of interdental papilla measurements.**

| Group | N | Papilla Height | Facial-lingual thickness | Interdental distance |
|---|---|---|---|---|
| Open Contact Point | 10 | 4.17 ± 0.51 | 8.70 ± 0.87 | 2.05 ± 0.55 |
| Complete Papilla | 110 | 3.99 ± 0.61 | 8.67 ± 0.92 | 1.81 ± 0.39[†] |
| Deficient Papilla | 15 | 3.99 ± 0.62 | 9.01 ± 0.81 | 2.07 ± 0.52[†] |

**Notes.**
All measurements are expressed as the mean ± standard deviation (mm).
Data with the same superscripts represent statistically significant differences, $p < 0.05$.

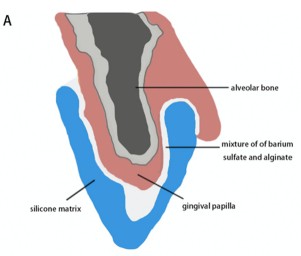 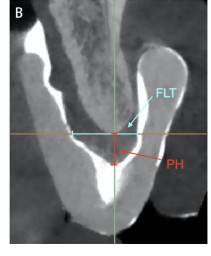 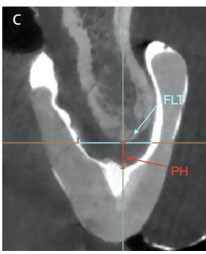 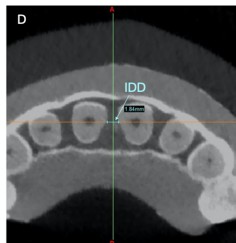

**Figure 2   The sagittal axis of cone beam computed tomography imaging measurements of height and width of the interdental papilla.** The sagittal and axial plane of cone beam computed tomography imaging measurements of PH, FLT and IDD. (A) Schematic drawing of cone beam computed tomography image to show the pyramid shape of the interdental papilla morphology using radiopaque impression technique. (B) The pyramid shape of the interdental papilla image and the measurement of PH and FLT in sagittal plane. (C) The col shape of the interdental papilla image and the measurement of PH and FLT in sagittal plane. (D) The measurement of IDD in axial plane. PH, papillae height; FLT, facial-lingual thickness; IDD, interdental distance.

gingival papillae in the anterior maxillary sextant were categorized: pyramid shape and col shape, with distribution of 97% and 3%, respectively (Fig. 2). The mean and SD of the PH, FLT, and IDD values obtained are listed in Table 1. The mean value of IDD obtained from the complete papilla group was significantly smaller than that of the deficient papilla group ($p < 0.05$), Table 1. In Table 2, the mean values of the PH at the proximal regions obtained range from 3.87 ± 0. 63 mm to 4.13 ± 0.56 mm. There were no statistically significant differences among the different sites ($p > 0.05$). The mean values of the FLT were statistically significant differences among the measured sites ($p < 0.05$). The mean value of the IDD between central incisors was significantly larger than the IDD values between central and lateral incisors ($p < 0.05$).

## DISCUSSION

This study described a gingival tissue measurement technique based on CBCT imaging, in which a mixture of opaque agent and alginate impression material was used to assist visualization of the gingival tissue. The CBCT images showed the bone crest which was the bottom of the IP and indirectly showed the profile of papillae aided by the opaque agent. The consistency of opaque agent mixed with alginate impression material maintains adequate viscosity and adaptation to the gingival surface and interdental papilla areas

**Table 2  Results of gingival tissue measurements at different interdental sites.**

| Measurement | Between central incisors | Between central and lateral incisors | Between lateral incisor and canine |
|---|---|---|---|
| Papilla Height | $4.13 \pm 0.56$ | $3.87 \pm 0.63$ | $4.07 \pm 0.58$ |
| Facial-Lingual Thickness | $9.37 \pm 0.88^{\dagger}$ | $8.20 \pm 0.76^{\dagger}$ | $8.89 \pm 0.77^{\dagger}$ |
| Interdental Distance | $2.04 \pm 0.50^{\ddagger}$ | $1.72 \pm 0.40^{\ddagger}$ | $1.91 \pm 0.38$ |

**Notes.**
All measurements are expressed as the mean $\pm$ standard deviation (mm).
Data with the same superscripts represent statistically significant differences, $p < 0.05$.

(*Cao et al., 2015*). The CBCT imaging revealed two types of interdental gingival contours: pyramid shape and col shape in maxillary anterior dentition, which was revealed by histological examination (*Cohen, 1959*; *Ko-Kimura et al., 2003*). The mean PH of different interproximal sites of central incisors, central and lateral incisors, lateral incisor and canine were $4.13 \pm 0.56$ mm, $3.87 \pm 0.63$ mm, $4.07 \pm 0.58$ mm, respectively, in accordance with other reports of the literature (*Chu et al., 2009*; *Nordland & Tarnow, 1998*; *Nozawa et al., 2011*; *Tarnow, Magner & Fletcher, 1992*). The mean PH of the different groups in the current study varied between 3.99 to 4.17 mm and the mean value was about 4 mm, in accordance with the two previous study (*Chen et al., 2010*; *Cho et al., 2006*) which reported that the papilla was present more frequently when the CP-BC distance was approximate 4 mm. The present data revealed that the PH may not be affected by the bone crest and contact point and the PH seems to be similar in height between complete and deficient papilla condition. The results led to accept the null hypothesis that no significant differences would be found among different conditions of tooth contact and presence of IP. The finding was a little different with a previous study (*De Santana, De Miranda & De Santana, 2017*), which founded that the dimensions of the IP are significantly increased in teeth without a contact point in comparison with their contra-lateral controls with a contact point in periodontally healthy subjects. The difference may be explained by the measurement method of PH, the present study measured the distance between the bone crest and papilla tip based on CBCT imaging, while *De Santana, De Miranda & De Santana (2017)* employed the horizontal line (ZL) connecting the gingival zenith from the adjacent teeth on silicon impressions as the base of IP. Despite the different results, the two study may present a new understanding of the role of the anatomic variables to determine the presence and size of the IP. As the dimension of interdental gingiva appears to be genetically determined (*Müller & Eger, 2002*). Furthermore, well designed, adequately controlled investigations in the subject are highly warranted.

The FLT of the papilla base was thicker in the deficient papilla group than in the open CP and complete papilla group (mean 9.01 mm versus 8.67 mm, 8.70 mm), but the differences were not statistically significant among the groups (Table 1). The results indicated that the FLT does not appear to have any effect on the papilla presence or PH. The findings were in accordance with a previous study which reported the measurement of the FLT of the papilla as the distance between the facial and lingual mid-point of a line connecting the most apical margins of the two maxillary central incisors in study cast (*Kim et al.,*

*2011*). The FLT assessment level at the proximal bone crest goes along with the proximal cement-enamel junctions of the maxillary anterior teeth (*Ash, 1993*). Consequently, the FLT was highly correlated with the facial-lingual dimension of the maxillary anterior teeth (Table 2).

The IDD of the papilla was significantly narrower in the complete papilla group than in the deficient papilla group with the mean values of 2.07 mm and 1.81 mm, respectively ($p < 0.05$) (Table 1). Results of some previous studies (*Cho et al., 2006*; *Martegani et al., 2007*; *Kim et al., 2011*; *Kolte, Kolte & Mishra, 2014*) suggested that the IDD is a significant predictor for papilla presence. However, data from other investigations reported that IDD was not a significant factor on multivariate analysis along with controlling for age and CP-BC (*Chang, 2008*; *Chow et al., 2010*; *Chen et al., 2010*). One study even found that there was no association between IDD and papilla presence (*Chow et al., 2010*). In fact, the effect of an increase in IDD on papilla presence became more prominent with increased CP-BC (*Kim et al., 2011*). The differences in periodontal status of the subjects and methods among these studies make direct comparisons difficult (*Chow et al., 2010*). Thus, The relationship of IDD and presence of IP needs further investigation. The IDD of the papilla between the central incisors was wider than that of the papilla between the central and lateral incisor ($p < 0.05$). Owing to the difference in the IDD of the papilla, the mesio-distal diameter of the crown portion of the maxillary central incisor is relatively wider than that of the maxillary lateral incisor (*Ash, 1993*).

Lack of volumetric calculation of the IP space was the limitation of this study. Another limitation is the small sample size that needs to be addressed and will be corrected in a future study to validate the findings and results.

## CONCLUSIONS

Within the limitations of this study, the results demonstrated that the CBCT with indirect soft tissue imaging technique is feasible to obtain a clear three-dimensional profile of the interproximal soft and hard tissue of maxillary anterior region. The conditions of tooth contact between upper anterior sextants may not influence the presence and dimension of the IP spatial structure in young Chinese adults with periodontally healthy gingival tissue. This finding may provide a new insight in the relationships of the interdental papilla with other anatomic factors for the diagnosis and treatment planning of clinical cases involving IP region.

## ACKNOWLEDGEMENTS

We would like to thank Dr. Dianne Gan for her help in preparing this manuscript.

### Funding

This study was supported in part by the National Natural Science Foundation of China (No. 61876005) and the Capital Medical Development and Research Fund, PRC (No.

2011-4025-04). The funders had no role in study design, data collection and analysis, decision to publish, or preparation of the manuscript.

### Grant Disclosures

The following grant information was disclosed by the authors:
National Natural Science Foundation of China: 61876005.
Capital Medical Development and Research Fund, PRC: 2011-4025-04.

### Competing Interests

The authors declare there are no competing interests.

### Author Contributions

- Gang Yang conceived and designed the experiments, performed the experiments, analyzed the data, prepared figures and/or tables, authored or reviewed drafts of the paper, and approved the final draft.
- Jie Cao conceived and designed the experiments, performed the experiments, analyzed the data, authored or reviewed drafts of the paper, and approved the final draft.
- Wenjie Hu and Kwok-Hung Chung conceived and designed the experiments, analyzed the data, prepared figures and/or tables, authored or reviewed drafts of the paper, and approved the final draft.

### Human Ethics

The following information was supplied relating to ethical approvals (i.e., approving body and any reference numbers):

Biomedical Ethics Committee of Peking University School of Stomatology approved this research (PKUSSIRB-2012047).

### Data Availability

The raw measurements are available in the Supplemental Files.

### Supplemental Information

Supplemental information for this article can be found online at http://dx.doi.org/10.7717/peerj.10006#supplemental-information.

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
