# Peer review of "Measurements of the gingival papillae architecture using cone-beam computed tomography in young Chinese adults"

_PeerJ, doi:10.7717/peerj.10006_

## Round 0.1 · original submission · Major Revisions

Although all three reviewers believe that the manuscript is a valuable addition to the literature, they also had various concerns. Two of the reviewers expressed a concern that the use of CBCT for the evaluation of the gingival papilla (rather than conventional techniques) is not justified sufficiently. When you resubmit the manuscript please be sure that you have addressed these and all of the other comments of the reviewers.

Reviewer 1 ·

Basic reporting

Yang’s paper explored some morphological parameters, including PH, FLT and IDD in maxillary anterior teeth by CBCT and provided anatomical data for aesthetic treatments. Its English expression was fluent.
Some minor points needed to be improved:
There were some researches concerning the direct measurement for PH and papillary width in mouth, such as “Analysis of the gingival biotype based on the measurement of the dentopapillary complex”, “The gingival biotype revisited: transparency of the periodontal probe through the gingival margin as a method to discriminate thin from thick gingiva”…….. Direct measurement in mouth is simple and noninvasive. What is the necessity and significance of the measurement using CBCT?

Experimental design

To measure PH, how to ensure the sagittal plane of each measurement was fixed? Was the sagittal plane perpendicular to the lip surface of the teeth or the line connecting center point of two adjacent root canals? For IDD measurement, there was also a same question.

Validity of the findings

no comment

Additional comments

Including samples in this experiment were young Chinese population. Are there any differences in PH, FLT and IDD in samples of different races and ages? It might be better to modify the title as “………in Young Chinese Populations”.

Reviewer 2 ·

Basic reporting

The quality of language is lacking for publication throughout. Both tenses and verbiage need improvement to make the text acceptable.
ie Line 1-20-Incomplete sentence
Line 25-Height of Papilla (PH)- simplified Papillae Height (PH)-correct tense
Line 44-45-According...present completely-improved by saying totally present or completely present
Line 51-The sentence should read- There is sparse "evidence" rather than investigation
Line 59-Contact "statuses"-no such word exists-perhaps "types" would have sufficed

Literature references are adequate- reference to Lyndon Cooper and his concepts on implants should have been included

Figures and tables adequate

Hypothesis and conclusions reasonable

The major concern is about the construction of the article. It confuses the topic rather than giving it clarification. The authors have done their due diligence in their investigation. They have failed to convey their findings in a cogent manner. A rewrite with the help of an editor with knowledge of the Gunning FOG index is in order. This will make the article readable and salient.

Experimental design

Well defined-see previous notes

Validity of the findings

Valid findings-concern should be discussed with regard to clinical relevance. Data is complete and statistically significant as noted. Do these findings really matter in the workplace at this time? Introspection is needed.

Additional comments

Marked improvement is necessary in writing to make this manuscript acceptable. The basic premise is reasonable. It must be readable.

Reviewer 3 ·

Basic reporting

1) The authors have made a commendable effort in writing a well referenced manuscript.

2) The language used in the article is clear and understandable.

Experimental design

1) In the present study the authors have utilized CBCT for assessing various gingival parameters for the presence and dimension of interdental papilla. However, in the segment of introduction (line no. 51-52) authors have noted "there is a sparse investigation on the relationship between the presence and the three-dimensional size of the gingival architecture (Chow et al., 2010)." this may not be completely rational, since there are several studies with conventional techniques that have tried to evaluate the same. Therefore it is suggested that the authors rephrase the given line.

2) CBCT is essentially related to the diagnosis of hard tissues. A precise and detailed justification for its usage, advantages over conventional techniques, drawbacks and clinical applicability in gingival analysis of the present study should be provided either in the segment of introduction or in the discussion.

3) In the clinical examination (Line no 80-83) the criteria for determining the presence or absence of papilla should have been more definitive. Authors would have benefited by utilizing the classification systems given by Norland and Tarnow or the Papilla Presence Index by Cardaropoli etc.

4) The CBCT examination is sufficiently detailed.

5) the measurement criteria of IDD (Line no 118-119) requires more description and clarity.

Validity of the findings

The conclusion drawn in the study regarding tooth contact and presence of IP require more explanation.

Additional comments

The authors are appreciated for a well conducted study. Though unrelated to the main manuscript, it would have been appreciated if an English translated copy of consent and IRB would have been provided.

---

## Round 0.2 · Minor Revisions

Please address the few remaining revisions suggested by the reviewers and return the manuscript to us. Thank you for re-submitting your paper.

Reviewer 1 ·

Basic reporting

Line 20 The aim of this study was to study……Two “studies” are used together, it is recommended to replace words.

Experimental design

no comments

Validity of the findings

no comments

Additional comments

1.Line 27 Statistical methods do not need to be included in the abstract.
2. It is recommended to indicate each reference surface and how to measure the parameters (PH, FLT, IDD…) in figure.

Reviewer 2 ·

Basic reporting

I have made correction to the manuscript and would like to send it as a PDF. Changes are made in red.

Experimental design

Design is reasonable. Other methodologies have been used in the past for these determinations

Validity of the findings

Statistical analysis is acceptable. A larger sampling (which the authors stated is planned) will give an improved power analysis.

Additional comments

Changes in syntax have improved readability for an international audience. Further changes have been done by the reviewer. These include reduction of compound sentences and tenses.

Annotated reviews are not available for download in order to protect the identity of reviewers who chose to remain anonymous.

---

## Round 0.3 · accepted · Accept

Thank you for making the requested changes and your patience with the reviewing process.